# Multi-site comparison of factors influencing progress of African insecticide testing facilities towards an international Quality Management System certification

**Sara Begg**[1]*, **Alex Wright**[2], **Graham Small**[3]*, **Matt Kirby**[2,4], **Sarah Moore**[5],
**Ben Koudou**[6], **William Kisinza**[7], **Diabate Abdoulaye**[8], **Jason Moore**[5], **Robert Malima**[7],
**Patrick Kija**[7], **Frank Mosha**[4], **Constant Edi**[6], **Imelda Bates**[1]

1 Liverpool School of Tropical Medicine, Pembroke Place, Liverpool, United Kingdom, 2 London School of
Hygiene and Tropical Medicine, Keppel St, Bloomsbury, London, United Kingdom, 3 Innovative Vector
Control Consortium, Pembroke Place, Liverpool, United Kingdom, 4 KCMUCo-PAMVERC, KCMUCo-
PAMVERC Test Facility, Moshi, Tanzania, 5 Ifakara Health Institute, Off Mlabani Passage, Ifakara,
Tanzania, 6 Centre Suisse de Recherches Scientifques en Côte D'Ivoire, Route de Dabou, Abidjan, Côte
d'Ivoire, 7 National Institute of Medical Research, Amani Centre, Amani Medical Research Centre, Muheza,
Tanzania, 8 Institut de Recherche en Sciences de la Santé, Bobo-Dioulasso, Bobo-Dioulasso, Burkina Faso,
Côte d'Ivoire

* sara.begg@lstmed.ac.uk (SB); graham.small@ivcc.com (GS)

doi.org/10.1371/journal.pone.0259849

TUNISIA

**Data Availability Statement:** Transcriptions of
interviews with facility staff are available from the
research group on request, on a case by case basis

## Abstract

### Background

Insecticidal mosquito vector control products are vital components of malaria control pro-
grammes. Test facilities are key in assessing the effectiveness of vector control products
against local mosquito populations, in environments where they will be used. Data from
these test facilities must be of a high quality to be accepted by regulatory authorities, includ-
ing the WHO Prequalification Team for vector control products. In 2013–4, seven insecticide
testing facilities across sub-Saharan Africa, with technical and financial support from Inno-
vative Vector Control Consortium (IVCC), began development and implementation of quality
management system compliant with the principles of Good Laboratory Practice (GLP) to
improve data quality and reliability.

### Methods and principle findings

We conducted semi-structured interviews, emails, and video-call interviews with individuals
at five test facilities engaged in the IVCC-supported programme and working towards or
having achieved GLP. We used framework analysis to identify and describe factors affeting
progress towards GLP. We found that eight factors were instrumental in progress, and that
test facilities had varying levels of control over these factors. They had high control over the
training programme, project planning, and senior leadership support; medium control over
infrastructure development, staff structure, and procurement; and low control over funding
the availability and accessibility of relevant expertise. Collaboration with IVCC and other

for the purpose of informing further research and on the condition that it will not be published in part or in entirety. They have not been made available as a dataset because they cannot be de-identified sufficiently to allow retention of anonymity. Interviews were conducted on the premise of anonymity, and this was the basis on which the study received approval from the LSTM REC. Requests for access to this data may be made to the research group at the Center for Capacity Research, ccr@lstmed.ac.uk, or to the LSTM Research Ethics Committee, rec@lstmed.ac.uk.

**Funding:** This work was supported by the Bill and Melinda Gates Foundation www.gatesfoundation. org [OPP1148615] awarded to IVCC. The funders had no role in study design, data collection and analysis, decision to publish, or preparation of the manuscript.

**Competing interests:** The authors have declared that no competing interests exist.

partners was key to overcoming the challenges associated with low and medium control factors.

## Conclusion

For partnership and consortia models of research capacity strengthening, test facilities can use their own internal resources to address identified high-control factors. Project plans should allow additional time for interaction with external agencies to address medium-control factors, and partners with access to expertise and funding should concentrate their efforts on supporting institutions to address low-control factors. In practice, this includes planning for financial sustainability at the outset, and acting to strengthen national and regional training capacity.

## Introduction

Insecticidal mosquito vector control products are key components of malaria control programmes in sub-Saharan Africa [1, 2], as well as providing the tools needed to eradicate malaria. With increasing insecticide resistance in malaria-endemic countries, new vector control products are urgently needed [2, 3], and these must be developed, tested and registered before they can be deployed. Vital to the testing of vector control products are the testing facilities in sub-Saharan Africa. These test facilities play a key role in assessing the effectiveness of vector control products against local mosquito populations in the environments where they will be used. The data generated during studies at these test facilities are used by manufacturers to support regulatory submissions for product approvals. Data need to be of a high quality to be accepted by these regulatory authorities. However, until recently, these test facilities either lacked or had inefficient formal quality management systems (QMS) to help manage data quality. Therefore, implementation of an appropriate internationally recognised QMS was required at these test facilities. After evaluating several QMSs, Good Laboratory Practice (GLP) was selected as being the most appropriate for the management of trials on vector control products. The purpose of a QMS compliant with GLP standards is to ensure that data generated during testing studies are reliably harmonised, repeatable and auditable. GLP is the gold standard for insecticide testing facilities and is advocated for by the World Health Organisation Prequalification Team for vector control products.

The Organisation for Economic Co-operation and Development (OECD) Principles of Good Laboratory Practice set the quality standards for the organisation and management of test facilities and for performing and reporting product testing studies. Data from GLP studies conducted in one OECD country are recognised by regulatory authorities in other OECD countries [4]. This mutual acceptance of data, and the cost and time savings associated with it, is a further driver for increasing the number of GLP certified laboratories with the capability of conducting laboratory and field studies on vector control products.

Laboratory capacity strengthening is an ongoing priority in low- and middle-income countries (LMICs), most typically as part of a health systems strengthening agenda. There is a good evidence base regarding the challenges and supporting factors that influence progress towards effective implementation of a QMS in clinical laboratories [5–11]. However, non-clinical laboratories have been less scrutinised. This is despite the role that these laboratories can play in public health [12], such as identifying new compounds or articles that may have use as a

therapy or for vector control and assessing the safety of newly identified compounds before they are used in clinical or human trials.

Increasingly, international collaborative partnership and consortia-based models are being used in the implementation of capacity strengthening programmes [13, 14]. To be systematic and sustainable, capacity strengthening must take an approach that responds to capacity needs at different levels (individual, organisational, and national/international) and be tailored to the context in which the capacity strengthening project is being implemented [15].

In 2013–4, seven insecticide testing facilities across sub-Saharan Africa were invited by the Innovative Vector Control Consortium (IVCC) [16] to join a programme of intensive financial and technical support for the development and implementation of QMS compliant with the principles of GLP and to achieve GLP certification. The purpose of this programme was to enhance the capacity of local research scientists to generate high quality and reliable field trials data across the region, with funding being provided by the Bill & Melinda Gates Foundation. These seven testing facilities form part of a larger network of trials facilities in Africa that help IVCC and its industrial partners to assess cross-resistance risk to new vector control compounds, and to evaluate the performance of prototype and final products.

The KCMUCo-PAMVERC test facility in Moshi, Tanzania was selected as a test case, to establish the feasibility of achieving GLP certification and succeeded in being granted certification in 2017 [17]. The experience and lessons learnt regarding the process of working towards GLP certification at KCMUCo-PAMVERC were useful subsequently to inform the joint approach by IVCC and the remaining collaborating test facilities. The purpose of the study reported here was to describe the factors influencing the rate of progress towards GLP compliance and certification across a range of these test facilities and identify lessons for future initiatives that centre around development of a sustainable laboratory QMS. In particular, this study sought to provide practical guidance for both individual test facilities on factors that may require particular or early attention, and for prospective consortia or partnerships on collaborative approaches to strengthening laboratory management capacity.

## Methods

### Study setting

Five of the seven test facilities supported by IVCC have been included in this study. These facilities are located in Tanzania, Côte D'Ivoire, and Burkina Faso. Facilities were selected for inclusion in the study based on diversity of context, encompassing test facilities from East and West Africa, and government, private, and Non-Governmental Organisation facilities. The key characteristics of each test facility are outlined in Table 1.

### Overview of approach for achieving GLP certification

Each test facility employed a broadly similar approach to moving towards GLP certification. Work began with a start-up meeting that included a test facility inspection by members of the IVCC GLP project team. The report from this inspection informed a needs assessment for the test facility, with a particular focus on infrastructure improvements and equipment procurement required to bring the test facility up to the standards required for GLP compliance. Each facility developed its own work plan covering construction, procurement, recruitment, documentation and training requirements for GLP, and budgets to cover the associated costs of plan implementation were reviewed and approved by IVCC.

The project was broken down into 3-month phases. Test facilities undertook work independently of IVCC support, working on all aspects of their work plan, and submitting progress reports and budget reports to IVCC on a quarterly basis. Progress was carefully documented,

**Table 1. Key characteristics of tests facilities selected for inclusion in this study.**

| Test Facility Name | Location | Government/Private/NGO | GLP status |
|---|---|---|---|
| KCMUCo-PAMVERC Test Facility | Moshi, Tanzania | Private | Certified 2017 |
| National Institute for Medical Research, Amani Center | Muheza, Tanzania | Government | Application for certification submitted 2019 |
| Centre Suisse de Recherches Scientifiques en Côte D'Ivoire | Abidjan, Côte D'Ivoire | Non-Governmental Organisation | Application for certification submitted 2019 |
| Institut de Recherche en Sciences de la Santé | Bobo-Dioulasso, Burkina Faso | Government | Working towards full GLP compliance |
| Ifakara Health Institute | Ifakara & Bagamoyo, Tanzania | Non-Governmental Organisation | Pre-inspection completed. Full inspection delayed by COVID |

and site visits undertaken regularly by members of the IVCC GLP project team. IVCC provided technical support and training opportunities, including training in computer systems and data management and in quality assurance monitoring for GLP compliance. Practice studies were used to identify non-conformances with the principles of GLP and to help target in-depth training of facility personnel. Construction, recruitment for key roles, and procurement for computer systems related to data management were early priorities, following learnings from the KCMUCo-PAMVERC test facility. Selected staff at each test facility also received business management and sustainability training, gaining insights into how to attract and retain business from companies wishing to conduct studies on their products, to secure the income necessary to sustain GLP certification in the longer term.

Once test facilities are confident in their documentation and facility readiness, an application for GLP certification is made to the South African National Accreditation System (SANAS) GLP compliance monitoring programme. This takes a phased approach to assessing the conformance of applicant test facilities to the principles of GLP. First, the test facility submits to SANAS copies of key documentation for review. Next, SANAS personnel conduct an on-site pre-inspection to assess the readiness of the test facility. Once the test facility has conducted at least one GLP study, SANAS inspectors conduct the initial inspection and study audit. This is followed by a non-conformance report and recommendation report. The test facility must address the non-conformances to the satisfaction of the inspectors, before documentation is submitted to the SANAS approval committee. Follow up inspections are conducted by SANAS at 6 months and annually thereafter to follow-up on any non-conformances raised at the initial inspection and to ensure that the QMS is still implemented and functioning properly.

## Study procedures

This study employed semi-structured interviews with staff at each of the five test facilities to construct a narrative of progress towards GLP compliance. A qualitative approach was used in order to capture the experiences and interpretation of those experiences by individuals in a wide range of roles [18]. This ensured that there was scope to describe both expected and unexpected events or factors, and offer explanations for these events or factors in order to identify lessons for future capacity strengthening initiatives. In-person interviews were conducted at three test facilities, whilst remote interviews were conducted at two test facilities due to travel restrictions resulting from the COVID-19 pandemic.

## Sampling method

A maximum-variation purposive sampling strategy [19] was used to select individuals for interview who had exposure to the GLP certification process, and included multiple

representatives from each level of a test facility to triangulate different data sources to determine the credibility and transferability of findings. Each test facility's organogram was used to identify relevant roles, with input from stakeholders at IVCC and the GLP manager at each test facility to identify individuals in each role.

## Data collection and analysis

Semi-structured interviews were conducted to explore the factors influencing the rate of progress towards sustainable GLP certification at their test facility. The interview topic guide was developed based on previous studies of laboratory capacity strengthening [10], with additional questions derived from findings from a case study of the GLP certification process at KCMUCo-PAMVERC [17]. Specific questions asked from the topic guide were matched to the roles and responsibilities of the interviewee. Interviews were audio-recorded and transcribed in full. All interviews were conducted by two researchers, one had a technical understanding of GLP requirements in insecticide testing facilities and the other had systems evaluation experience. In person interviews were conducted on-site, in private locations within the test facility.

A framework analysis [20] was used to identify themes emerging from the interview transcripts following the five-step process of familiarization, identification of thematic framework, indexing, charting, and mapping/interpretation. The framework approach to analysis was employed to facilitate identification and understanding of the interactions and processes that influenced the rate of progress towards sustainable GLP certification and to facilitate comparison between test facilities, offering a systematic approach to manage qualitative data [20] that allows for both deductive and inductive approaches to analysis. An initial coding framework was based on findings from the case study exploring progress at the KCMUCo-PAMVERC Test Facility [17], with codes based on challenges and enablers to progress identified at this site. Transcripts of interviews conducted in this study were read in detail. Following this familiarization with the interview data, these codes were revised, and further codes were identified and incorporated into the framework. At this point, emergent themes related to the degree of control test facilities had over the factors affective progress were identified, and this was used to structure the thematic framework.

This initially deductive approach, using findings from the KCMUCo-PAMVERC Test Facility to inform the initial framework, followed by an inductive approach to develop this framework, supported focus on the specific area of factors affecting GLP certification progress while allowing for unexpected aspects of the GLP certification process described by participants to be included in the analysis [18, 21], sharing some characteristics with Grounded Theory [22]. All interview transcripts were indexed with these revised codes, using NVivo software (QSR International). To identify sections of the data that corresponded to the relevant theme, a narrative was constructed to define the key issues, find associations between those issues and, where possible, provide explanations. In sum, this facilitated the development of a model to describe this qualitative data, which can be tested in future research [23]. Statistical and quantitative methods were not used in this study as test facilities were at substantially different stages in the process of working towards GLP certification, limiting the usefulness of comparison of potentially relevant process-based quantitative data such as checklists and rendering comparison of outcome indicators such as efficiency or data quality not possible.

## Ethical approval

Ethical approval to conduct this research study was obtained from the Liverpool School of Tropical Medicine Research Ethics Committee (approval number 18–041), the National

Institute for Medical Research Tanzania (ref NIMR/HQ/R.8c/Vol./I/554), and the Centre Suisse de Recherches Scientifiques en Côte d'Ivoire Institute Review Board (ref 19–549). In line with LSTM's guidance on Network and Capacity Strengthening Studies, senior management at the IHI (Tanzania) and IRSS (Burkina Faso) test facilities provided an institutional approval document to confirm willingness to participate and that the research did not require in-country approval.

Participants were informed about the research using participant information sheets. Written consent was obtained from each participant prior to participation in interviews.

## Results

A total of 65 members of staff from the five test facilities participated in this study: 66 members of staff were invited to participate, with one declining. Of these staff, 16 were laboratory/insectary technicians or attendants, 17 were from non-scientific administration/information technology positions, 22 were from scientific middle-management positions, and 11 were from scientific senior management positions. Forty-nine participants were male and 16 were female. Quotes from transcripts to illustrate key points are referenced throughout these results: quotes have been anonymised, highlighting the role of the interview participant but not their test facility.

From the interviews, eight factors were identified that substantially influenced the progress of test facilities towards GLP certification (Table 2). These were organised according to the degree of control (high/medium/low) that the test facilities had over these factors as it was clear from the narratives that the degree of control had major implications for planning for the sustainability of the capacity strengthening programme.

### Factors over which test facilities had a high degree of control

High-control factors that influenced the progress of test facilities towards GLP were related to organisation of existing assets and communication within the test facility.

**Training programme.** Test facilities implemented extensive internal training programmes and engaged a range of external specialists, encompassing a wide range of topics (Table 3). Training was regularly identified as a key factor with recommendations made for earlier and more detailed training:

**Table 2. Factors influencing test facilities' progress towards GLP certification.**

| High control | | Medium control | | Low control | |
|---|---|---|---|---|---|
| **Factor** | **Description** | **Factor** | **Description** | **Factor** | **Description** |
| Training programme | *Internally delivered training programmes and accessing externally delivered training.* | Infrastructure development | *Negotiation of construction rights, supervision of construction, and sourcing of materials.* | Funding | *Availability and accessibility of capital investment in research capacity strengtheing for insecticide test facilities.* |
| Project planning | *Use of meetings, working groups, time management and planning of order of project activities.* | Staff structure | *Organisation of personnel and recruitment of technicians, government mandated or otherwise.* | Availability and accessibility of expertise | *Availability of expertise related to: GLP, quality assurance, information technology, data management, calibration, maintenance, insecticide tests, and the extent to which these can be accessed by the test facility.* |
| Senior leadership support | *Senior leadership understanding and support of GLP purpose, value and requirements.* | Procurement | *Use of local and international providers, via internally or externally mandated systems.* | | |

*"If I were to start that process now, I will start with training. I would wish that we could be given no money but training, be trained a lot and then we start."*

Laboratory supervisor

Internally delivered training included general GLP workshops, procedure-specific training linked to Standard Operating Procedures (SOPs), and refresher training on topics including quality assurance and computer use. General GLP workshops, particularly those delivered on a regular basis, were well received by staff at all levels and contributed to a consistent and shared understanding of the GLP project:

*"I find it very useful because [GLP Manager], actually, helped us to change their mind; because to move on with this facility, people need to change the mind. Previously, we were running our business as usual but now we have to change, to be serious, to work hard, and follow the procedures, the SOP for running, for planning different activities."*

Data Manager

Awareness of why the test facility was seeking GLP certification, and understanding the personal benefits of working in a GLP test facility (including: job security, attracting new studies, a more comfortable working environment), increased buy-in to the project and willingness to adopt new working practices:

*". . . we did some introduction, training and explaining to them the advantage of the GLP, so everybody was like committed to make sure that we are into it . . . That training initiative helped a lot. If it were not for that, because they might think that we're just disturbing them to do something else."*

Study Director

Generalised GLP training with all members of staff also promoted engagement. For example, because facility drivers understood the GLP project, they took care to ensure samples were delivered in good time from field sites to central laboratories:

*". . .for example, sometimes you can think maybe the driver is not a part of GLP. Now we know even the driver is part of GLP. For example, we want to take the specimen from one*

**Table 3. Cited examples of training included as part of the GLP project.**

| Quality Management Systems | Science-Specific |
|---|---|
| • GLP principles training<br>• Internal training in Standard Operating Procedures<br>• Quality assurance for GLP<br>• Annual quality assurance refresher course<br>• Archivist training<br>• Good Clinical Practice training<br>• Data Management for GLP | • Real Time QPCR training<br>• Animal ethics<br>• Mosquito biology<br>• Malaria workshops<br>• Animal keeping<br>• Small animal sedation |
| Safety | Other |
| • Health and Safety training<br>• Fire training<br>• Risk Assessment training<br>• OSHA training in chemical handling<br>• TPRI (Tropical Pesticides Research Institute) training | • Business sustainability workshops<br>• External training in leadership/management<br>• Communication with management training<br>• Internal training in computer use<br>• Financial management |

*place to another and if the driver he or she don't know what is GLP, he come late and then you have your specimen and the time is still going. So, your specimen is not good. Now all of us we are all ready, even the drivers they know. The administration they know."*

Technician

However, sustaining the intensity of training was a challenge:

*"It's about two months we haven't met. It was weekly."*

Human Resource Manager

Whilst generalised GLP training supported facility-wide buy in and support for the project, role-specific training was important in ensuring that staff understood their individual role in the context of GLP:

*"What is required, a lot of information, but they were for different—Because we are different people. Because the information for Study Director, that a Study Director should make sure— You must do this, this, this. The person who's doing the experiment must do this, this, this. The person who's archiving the data—It was a whole package, but it gave us a general picture of—Every person knew what he's supposed to do in this."*

Quality Assurance Manager

The most consistently cited challenge with internal training programmes was the amount of ongoing on-the-job training and supervision needed to support technicians to change how they conducted and documented procedures. In some cases, this included asking technicians to change habits of more than 10 years:

*"One thing is that when you come with new things, it means that you have to change the habit of people, which is not easy because people have their way to do and you say no. I know that you are doing like this they want to improve what you are doing. It means that if they have already a form to maybe evaluate the maybe record temperature or they have a form for something else, you have to say to them no. The SOP, this is the way we have to do the form. Sometimes they have problem. Where they say 'No. We used to use this form more than 10 years, why you are asking us that this is not okay?' It takes time to explain."*

GLP Manager

Practical approaches to training, including regular practice, helped technicians develop new habits. As observed by a technician:

*"The way he explains things. He will sometimes ask you like, 'Why do you do this? Why do you cut the ends of the pipette?' He was asking us, 'Are you doing that just for routine, or there's a reason for that?' Then I will say, 'According to the SOP, you have to make the hole bigger, so that the tube will not get stuck.' He was giving us questions, quizzes, every day, until he was comfortable that we're good".*

A combined practical training, SOP review, and formative assessment process used in some test facilities engaged technicians when changing procedures and documentation. This helped

ease this difficult transition period while also streamlining the process of developing SOPs and documenting training and competency in techniques:

> *"The training was the beginning before we start each and everything, we were trained and then we work practically with [IVCC staff member]... we enjoyed it because we are not learning in theory, we both theory and practicals, so we read and do it practically."*

Technician

Externally delivered training included formal and informal support by IVCC on topics including GLP principles, SOP development and implementation, techniques, quality assurance and data management. This was delivered in person at the test facility, or remotely via teleconference and email:

> *"Some of the things that we were working with, we had a lot of communication with [IVCC staff], 'We're stuck here,' telephone conference and things like that."*

Laboratory Supervisor

This was identified as beneficial for overcoming sticking-points and facilitated further inter-facility learning as IVCC staff acted as a central resource. The network of collaborating test facilities was particularly mobilised for inter-facility learning for data management and IT infrastructure, building on the experiences of KCMUCo-PAMVERC:

> *"No, actually, previous I was doing these things but I was I was not doing at the level that is required by GLP. I have already attended one training which was conducted in the KCMC, Moshi. That one has actually changed a lot of things in my mind about data management and documentation."*

Data Manager

Attending international training related to GLP principles and sustainability for GLP was made difficult due to visa issues. This was easier for travel within Africa:

> *"No, within the African continent it's very easy, yes. It's much easier even to go to Asia it's much easier... but to go to UK that is a challenge."*

Facility Director

Although there were still incidents of visa problems travelling between West and East Africa:

> *"With that, I think last year, I prepared to go to Tanzania to do data management training and I had a problem for visa. I didn't move there."*

Data Manager

Beyond training for new or existing staff, consultants were an effective means of bolstering a test facility's capacity on a temporary basis, and consultants were used in particular for GLP project management, auditing, and quality assurance:

*"[On a consultant] Previously, we did not know exactly how to put in place. He helps us to, let's say, to coordinate everything together because we know the different activities that we need to do. We do not know how to plan, for example to collect our equipment to other equipment for the calibration. I do not know exactly. He helps us to connect our facility here, our equipment with other equipment for the calibration and so on."*

Human Resource Manager

Identifying and engaging appropriate external consultants and training providers could be a slow process, due to a lack of regional providers. IVCC was able to provide some training, and as a result of the partnership between the test facility and IVCC, it was possible to address this challenge more effectively. This was, nevertheless, linked to the low control factor "availability and accessibility of support and expertise".

**Project planning.** Effective project planning was a key factor in overcoming the delays and challenges faced by test facilities. Project plans were put under pressure by unexpected disruptions, including flooding and changes in senior management, for example:

*"Yes, there were delays during the heavy rains... We sometimes have a lot of rains. The road is not very friendly during the heavy rains, the period. It was not possible for heavy trucks to bring there some materials for construction. That was major challenge."*

Laboratory Supervisor

At some test facilities, this limited the number of tests they were able to include in their application to SANAS for GLP certification.

Regular meetings to review progress, highlight successes (both individual and institutional) and discuss challenges as a group were beneficial across test facilities where they were implemented. For management staff, these helped track project progress and sustain momentum. For technical and auxiliary staff, these meetings were highly motivating because they recognised individual and group efforts and reinforced the collective, team effort required to achieve GLP:

*"Changing old habits is not easy thing. To overcome that issue, monthly meetings that we undertook were useful, as during those meeting, that issue was discussed and the necessity of GLP certification was reminded with rigor and insistence."*

Laboratory Supervisor

At one test facility, where technicians did not always feel able to air their views through team meetings, a system of technician and laboratory supervisor meetings feeding into senior management via an intermediary GLP manager allowed issues to be discussed and solutions identified. Working groups addressing specific areas of the GLP project were effective both in ensuring progress on that area and promoting ownership of the progress:

*"The challenge we were facing with those new infrastructures was the respect of the recommendations by the builders. For this reason, in addition to the administrative and technical supervision team usually established for that kind of building, we have established our own supervision team. The different teams were meeting weekly with the aim for our team to make sure that everything was being done according the recommendations and GLP requirements."*

Laboratory Supervisor

Examples of working groups included overall GLP project management, construction, health and safety, and training programmes. These working groups included representatives from all levels within the test facility.

*"On my side, technical aspect of the GLP, we also have a team which is actually led by me, [GLP Manager], and two more people. We are five altogether. Even [technician] is in that team. We have also the person from the accounts section and also from the procurement unit is also involved. We are actually reviewing weekly trying to see how these projects are going on, what are the challenges? Is there a need of maybe arranging a meeting with the director or somebody else about, do we need to sit with the contractor to rectify? Although it was a weekly meeting, but it was like every day."*

Administrator

The order in which activities were addressed allowed test facilities to tackle some issues in parallel. In particular, while infrastructure development, construction and rehabilitation was the activity that was implemented first at all test facilities, and was viewed as being the most important thing to address early on, other activities could run alongside this, including: recruitment of key roles (particularly quality assurance and data management); procurement (particularly of hardware and software necessary for computerised data management systems); and development of documentation related to the wider institution systems such as procurement and human resources:

*"When we have been doing this renovation, there is other activities which has been going on like preparation like organogram, ordering of some of the equipments, laboratory equipments. The preparation of staff on the training about the GLP."*

GLP Manager

The work required to prepare and implement documentation for GLP, such as developing and checking SOPs, was substantial at all test facilities. In many cases, test facilities were working from a baseline of very few SOPs. SOP development had to be undertaken alongside routine funded project work putting a substantial burden on technicians:

*". . . we need to write the SOP with our technician who have been using all the SOPs but the number of technicians in our institution is very small, so we are not available in developing SOPs."*

Study Director

Individuals in other roles also felt this strain on their time, with study directors having to dedicate significant amounts of time to SOP development to alleviate pressure on technicians:

*". . .developing SOPs is a massive, massive role. [Study Director] and I get almost no writing done apart from writing SOPs for years, at least a year and a bit."*

Study Director

In some test facilities, these pressures were eased by appointing additional staff, often funded by IVCC and is, therefore, linked to the low control factor "funding".

**Senior leadership support.** GLP project staff valued senior leaders who understood the purpose of GLP and recognised the future benefit to the test facility in terms of data quality, prestige, and the ability to attract new studies. Senior leadership understanding and support of the GLP project was essential for progress. For example, delays in signing off budgets were minimised, and therefore barriers to progress on construction, procurement, and recruitment were reduced. In one test facility a change in leadership facilitated a rapid acceleration in progress as the incoming director of the test facility had been previously engaged in the GLP project. In another, a senior manager's attendance at a workshop on GLP sustainability resulted in increased engagement and support for the project:

> "*Even this boss that we have was not in such a passion in the beginning than he is now after the training which they were taken, I think, to Liverpool, something like that. It was a training on business-related issues.*"

GLP Manager

## Factors over which test facilities had a medium degree of control

Medium control factors that influenced test facilities' progress towards GLP were more complex than high control factors and were characterised by interactions with systems relatively close to the test facility, such as host institutions or government systems.

**Infrastructure development.** The rehabilitation of existing buildings, construction of new buildings, and installation of necessary utilities and equipment was extensive at all test facilities (Example infrastructure improvements, Box 1). This was typically the first group of activities to be undertaken. The construction works required varied across test facilities, necessitating bespoke plans developed between IVCC and the test facility leadership. Developing the physical infrastructure for GLP was bureaucratic, costly and time-consuming:

> "*We needed to build a new laboratory. It took a long time to raise the funds necessary for construction. Also, the space to build the laboratory was difficult to obtain. Several discussions with the administration and IVCC have made it possible to build a new laboratory.*"

Laboratory Supervisor

---

### Box 1. Example infrastructure improvements, drawn from work at Ifakara Health Institute

- New build animal house

- New field study bed net washing plant with 2 x 100,000L evaporation tanks

- Purchased three 40 ft shipping containers, built a roof over them and converted two for storage and laboratory IRS study block rooms

- Refurbished laboratory bed net wash, testing lab and holding room

- Refurbished old insectary building

- Refurbished field station with environmentally controlled rooms, new tables and plastic sterile wall cladding

---

- Purchased land for experimental hut site

- Increased capacity of huts from 10 to 28 with concrete platforms built for all huts with insect channels

**Note:** Funding for infrastructural improvements at Ifkara Health Institute was made possible through seed funding for infrastructure from IVCC. These investments attracted further projects, generating funds for further investment in the facility's infrastructure.

At some test facilities, additional investment was required to install utilities and repair roads:

"... in [Field Study Site] there was no house, so he was obliged to go to great lengths and those lengths was very far from the town. That means you have to set up electricity, water, let's say water sources. This also was a great challenge for us."

GLP Manager

At four test facilities, issues of land ownership influenced progress on infrastructure rehabilitation and construction. If test facilities did not own the land on which they planned to build, this caused disruption to progress on the project, either because it required negotiation with the landowners, or because project costs were inflated as test facilities purchased the land needed:

"It took a little time for us to get the initial approval not from [Parent Institution] but from the board that we'll be allowed to make any changes that direct any structure around. That took time to negotiate, but when it was finally agreed then I was free to construct a structure."

Test Facility Manager

Cost saving measures included test facility staff taking on construction project management roles, led by a working group drawn from laboratory staff and administrators. This came at the cost of their own time but facilitated local sourcing of construction materials and laboratory fittings, reducing overall costs:

"I had to use my brain to see how best we can do it. The idea which came into my mind was that maybe we invite these local contractors, but we need to supervise them. This was a huge task. You invite a contractor and then you buy all the materials. What you want to do is to borrow his labour charges. You procure all the materials. This is an area also which was a bit tricky because you had to have someone going around shop after shop, the hardware shops"

GLP Manager

Test facilities were also able to use the investment by IVCC to secure match funding from government and other partners towards construction of facilities:

"I can say that actually even 50% of the whatever we have achieved from other sources apart from IVCC, but the initiator was the IVCC. It has opened our minds that we can do that. If

*that is why the other people chipped in like the government, the institutions. They committed indeed."*

Test Facility Manager

This was made easier as visible progress began to be made on the facility.

Test facilities in Tanzania had the opportunity to learn from KCMUCo-PAMVERC. This saved time that they might otherwise have been spent interpreting the GLP principles and applying them to the facilities they were developing:

*"First of all, we had a road map. Our experimental huts were old, and they need to be reno-vated. We visited the other site to see how they have done it. We had a model to see exactly what we are supposed to- where are we supposed to go from where we are."*

GLP Manager

There was a clear appetite amongst West African test facilities to similarly learn from test facilities that were further along the path to GLP certification:

*"For example, in [Collaborating Test Facility]. I think they're GLP now, so as it's not far we can organize a touring there to see how they manage their sites."*

Laboratory Supervisor

**Staff structure.** The structure of staffing at test facilities influenced the extent to which test facilities could reliably staff GLP studies. In particular, the organisation of technicians and how these roles were filled required specific attention.

In one test facility, most of the work undertaken at the test facility was by external research-ers and PhD students, with only a small number of full-time technical staff to support this work:

*"I'm the only technician here doing my work, but in terms of GLP everything has become tough. No access to do things because I'm the only one, so we will need maybe more personnel, more technicians."*

Technician

This put an undue burden on these technicians throughout the GLP preparation process and meant that the permanent workforce would not be sufficient for the conduct of a fully GLP compliant study for industry. It also increased the training and supervision burden on the technicians to ensure that visiting researchers complied with GLP requirements (e.g. using personal protective equipment).

In the government run test facility, staff could be moved or given additional assignments at short notice:

*". . . in GLP, we need someone to stick at the working place. If you are in the ITF, you are in the ITF, you should stick there. But as a government employee, sometimes you're needed to go somewhere for a certain activity. I think that is another challenge. This is different from non-governmental institutions."*

Laboratory Supervisor

As a result, all roles needed a deputy to minimise disruption to the GLP project. The additional staff were not funded by the government so, for all facilities, it was imperative to attract studies from industry to cover additional staff costs. Interviewees indicated that it was not unusual for individuals in senior scientific roles to work more than full-time in order to make progress on GLP certification:

*"...we subsidise research by working more than 100% FTE [Full Time Equivalent] with the Test Facility Manager and one Study Director paid from projects outside of the unit."*

Test Facility Manager

**Procurement.**   Delays and long lead times, particularly for international procurement, meant that procurement of both one-off purchases and consumables for studies was a key factor in test facilities' ability to make progress with GLP certification:

*"Yes. Some products are not easy to find in our area. It's not easy to find some laboratory product in [country] and then when you find it here, it's very expensive. Then [GLP project manager] can say okay, I will take it in London, take it in London is quickly where we put it on the boat one month or two weeks for example, and then they call you to say–ah, the material is here. We don't pay VAT because we are an international organisation, we don't pay VAT in [country], and then ask for exoneration from the Ministry of Foreign Affairs, we ask them to send us an exoneration document that can take 1 to 10 days. That's why the delay can be very long."*

Procurement Officer

These delays originated from both internal and external systems. Internally, excessively bureaucratic systems and slow action by procurement departments, as well as delays in sign-off on purchases by senior management, were the principle sources of delay. Most test facilities included the streamlining of internal procurement processes as part of their GLP project. Engagement of the procurement and accounts departments and institutional leadership in generalised GLP training improved support of these changes:

*"...the good lab practice to us as an accountant, we're a little bit far from the lab but we are part of them, we're doing our part of finance which assists the good lab practice. If they need items on time, item from the store, yes, we're trying to push them to push those lists on time."*

Accountant

Test facility independence determined how easily adjustments could be made to procurement processes, with particular challenges faced by institutions linked to government or wider networks of institutions:

*"Yes, maybe I can take you to experience the way I work with these guys here. They can have a test in the lab inside, and sometimes, they fail to ask that this component or chemical, they won't be enough for us to facilitate this test in the lab. When they find that the things are not enough while they're inside the test, they come and ask us, 'Can you go and buy it for us fast, maybe fast-fast?' Here, we have the process because we need to fill the document that you need the items and we need to take that document to head of department. Then from the head of the department to the center director or PI, we call it PI. Then from the PI to accounts department. Then from accounts department to cashier."*

Accountant

Shipping times and delays to delivery when goods were held by customs were also factors that influenced progress. Local rather than international procurement was prioritised at many test facilities, reducing lead times, but risking lower or more variable quality products. Procurement of key international purchases related to computer systems, such as servers, were an early priority at test facilities.

## Factors over which test facilities had a low degree of control

Low control factors that influenced test facilities' progress towards GLP were systems-level or environmental in character.

**Expertise and support.** GLP-specific expertise often had to be sourced from outside the test facility, through recruitment, use of consultants or training of existing staff. The local or regional availability of appropriate providers of this training varied between test facilities, with some test facilities facing delays to progress while trying to source an appropriate provider, especially for general GLP training, and training for quality assurance and data management roles.

Data management support was made possible as the IVCC GLP project team developed their own expertise in data management for GLP as the project progressed, through a continuous learning process. This knowledge was then disseminated through the collaborating test facilities. Furthermore, all test facilities benefited from data management training for GLP from the data manager at KCMUCo-PAMVERC:

*"Yes, for example, right after coming back from the training, from Moshi, there was one project that was going on. I started doing those small procedures in how to manage data and how to collect the data in a proper mechanism. Also, doing, for example, in that training also they insisted how to use the double-entry, which you are not doing, but now we are doing four projects doing the double entry."*

Data Manager

Calibration and maintenance of equipment was typically provided by a range of external providers, although finding providers who were accredited to do the work to the GLP requirements was a challenge, reflecting a lack of regional expertise and capacity:

*"There are no accrediting body in [Country] which is a problem. In the maintenance of the equipment, the suppliers often don't have the technology base."*

GLP Consultant

As well as calibration and maintenance of equipment, test facilities required support from government bodies that oversaw waste management and animal welfare. As with calibration and maintenance of equipment, these were services and support systems that test facilities needed but had very little control over.

**Funding.** The implementation of the GLP project was costly with an average test facility budget of nearly $250,000. Additional costs to test facilities outside of the immediate GLP project increased the financial burden on test facilities. In particular, there were costs associated with meeting some GLP requirements in non-GLP studies, in order to maintain a GLP compliant environment:

*". . . on average that a GLP lab has a 30% increase in costs compared to a non-GLP lab. I think we pretty much have seen that over the past couple of years."*

Study Director

The costs associated with sustaining the GLP system beyond the period of investment by IVCC was a real concern for test facilities, with increased staff costs and increased overheads from running systems such as air conditioning units or generators:

*"Although you might seek initial grants to establish the GLP, but sustaining it is a very big challenge. For example, you might be having a grant to enable you to have a power backup, which is a big problem in Africa to have reliable electricity. That generator, over the time, will require services and probably replacement later and that is costly. Sustaining cost is something to be expected. Not only what the initial cost is, but also you have to think about how are you going to sustain it."*

GLP Manager

Despite a good understanding of the business case for GLP and the need to attract studies from industry, test facilities generally felt they did not have the marketing skills necessary to sell their services on the open market, and identified this as a major risk to sustaining GLP certification:

*". . .it should be continued particularly on this marketing actually. Because now we're going on marketing competitions. The GLP site should be trained on marketing. Marketing themselves with marketing their services, what they have, and also negotiation skills, and on contractual issues"*

Test Facility Manager

Furthermore, test facilities were concerned that there was a disconnect between the sharing of best practice between test facilities and competing for studies as a means of ensuing sustained GLP certification, and that some test facilities had a competitive advantage through factors such as subsidised staff costs that meant they could offer studies for less than cost-price.

A degree of flexibility from IVCC on funding amounts, to pick up shortfalls in budget when unexpected costs arose, as well as match funding provided by, for example, the government or other institutions, was key for several test facilities. In particular, this ensured construction projects could be completed:

*"This is the lesson what I've learned here because I know that we have got a lot—financial support from IVCC. Actually, that was not enough to achieve what we have now. I can say that actually even 50% of whatever we have achieved from other sources apart from IVCC, but the initiator was the IVCC. It has opened our minds that we can do that. If that is why the other people chipped in like the government, the institutions. They committed indeed."*

Test Facility Manager

This allowed test facilities to allocate dedicated time to working on the GLP project rather than in addition to full-time work on projects:

*"I think for me first of all, IVCC did a lot of effort to extend the contract. That was for me the biggest support which helped us to make this big jump. Definitely, and that helps us now to recruit two people who were really key to help us to go."*

Director of Research

Test facilities which were not able to access this match funding found that progress was delayed as they were required to maintain a full schedule of studies throughout the process in order to cover core costs including institutional overheads, and therefore had less time to complete additional work needed to implement GLP:

*"It had been difficult especially prioritising the GLP project while there is limited fund to sustain personnel employment and facility business. Therefore, we were balancing between GLP project activities and other projects that was meant to generate some funds to support GLP infrastructure/facility improvement."*

GLP Manager

## Discussion

Eight major interacting factors have been identified that influenced progress towards GLP in these five test facilities (Table 2). The organisation of factors into high, medium and low control categories reflects the perspective of the test facilities and test facility staff. High control factors were largely associated with organisation of existing resources within the test facility, mobilising knowledge, competencies, and channels of communication to optimal effect to deliver training programmes, implement a project planning approach, and engage senior leaders within the test facility. In contrast, low control factors were located far from the institution and individuals within the institution, driven by the regional or international availability of key resources. Medium control factors sat somewhere between the two and were often associated with interactions with external agencies and systems relatively close to the test facility, such as parent institutions or government systems. These distinctions highlight where the roles and responsibilities of partner organisations in research capacity strengthening projects might lie. In this case, these test facilities, in partnership with IVCC, were able to successfully negotiate the low control factors. This was largely possible because IVCC had access to funding and expertise that could overcome challenges related to these factors. Funding and expertise were, from IVCC's perspective, high control factors. As a result, the principle role for IVCC was to address the low control factors by providing funding and facilitating access to expertise and support, while the role for the institution was to address the high and medium control factors. In order to successfully make progress towards certification, therefore, action must consider all factors, and employ a systems approach to GLP certification projects. Below, we briefly discuss each control level, highlight relationships between factors and consider the role of collaboration in addressing these factors.

### High control

Senior leadership support was a highly interlinked factor in progress towards GLP certification. Senior leadership in test facilities can act to influence progress on multiple factors including infrastructure development, staff structure, procurement and funding. Senior leaders can also seek match funding on projects and draw on relationships with umbrella institutions to secure permission for construction, appointment of new roles, and implementation of new

procurement policies. Competencies identified for laboratory leaders [24] that would support all of these roles include those related to leadership, management, and communication, underpinned by responsiveness to minimise delays in progress driven by other staff within the laboratory. Engaging senior leadership such as directors early in projects helps support smoother progress. Strategies to facilitate this include specific engagement on the benefits of the quality management system, including business opportunities, staff motivation and prestige. Including senior leadership support as a selection criterion for inclusion as a collaborating test facility could promote progress on projects in the future.

Training programmes facilitate progress towards certification both by supporting development of essential knowledge and competencies and by promoting a shared understanding of roles, responsibilities and the purpose of the GLP certification project. We found that for the substantial amount of training that could be delivered internally, training that was regular, practical and highlighted the benefits of GLP certification served to reduce errors, building new habits, and increasing engagement, reflecting experiences in other capacity strengthening programmes [25–30]. Specific GLP training could only be provided internally once some staff had undertaken IVCC-funded external training, particularly on SOP development, data management, GLP principles and quality assurance. This consequently contributed to test facilities' "capacity to build capacity" [31, 32]. With multiple test facilities included in this programme there came the beginning of a regional network of trainers in GLP data management which may help facilitate staff retention post certification [29, 33].

Project planning and tracking utilised a range of internal audits, meetings, workshops, and working groups, all of which are useful for implementing a quality management system [25, 28, 29, 34, 35]. The principle barrier to effective project planning was staff availability, linked to staff structure and funding. A strong recommendation from study participants was to allocate budget to allow staff to dedicate time to planning, learning new skills and implementing activities.

## Medium control

Procurement, staff structure and infrastructure were all factors over which individual test facilities had a medium degree of control. Procurement is a widely recognised challenge in capacity strengthening [25, 36, 37]. Long lead times and complex, bureaucratic importing processes account for many delays, but internal bureaucracy and inadequate communication also contribute to procurement delays. Simplifying internal processes, prioritising local procurement, and senior leadership advocacy for simpler importing processes can minimise these delays.

Staff structure was often determined by the funding model of the institution, which ranged from government-funded positions, through roles subsidised by partner institutions, to institutions that had to cover the cost of all posts from delivery of studies for industry. Key supplementary roles such as coordinators or managers for the GLP project, which have previously been identified as vital in accreditation projects [38], were made possible through appointments funded by IVCC, while senior leaders were key for ensuring these posts were implemented.

Infrastructure developments that include construction, rehabilitation, and refitting of laboratory and office spaces are a necessary, extensive, and costly part of most laboratory accreditation projects, particularly for laboratories in LMICs which may be operating from a lower baseline [39, 40]. Securing permission to build on land or purchasing land to build on was also a recurring challenge across multiple test facilities.

## Low control

The availability and accessibility of expertise and support necessary for completing the GLP project was largely beyond the direct control of the individual test facilities, reflecting the

regional availability and system-level deficits. These were particularly related to training providers, and maintenance and calibration of equipment. One of IVCC's principle roles within each project was to fill the gap in regional training providers, by providing tailored, locally relevant and accessible support and expertise, and by facilitating access to international training through funding. Maintenance of laboratory equipment has been identified previously as a bottleneck for strengthening health systems, with development of local capacity for maintenance identified as a key solution [41]. The rigorous requirements of GLP, that equipment be calibrated by an accredited service provider, make this need for development of maintenance and calibration capacity all the more pressing.

Funding interacted with all factors described above, directly or indirectly. For example, project planning was enhanced by a supportive staff structure, which was made possible through adequate non-study funding. Ultimately, while many test facilities had resources that could be organised effectively to allow progress towards GLP certification, funding facilitated this organisation while also allowing for investment in high-cost activities such as construction or international training. The second key role of IVCC within this project was, therefore, providing part-funding for the work required. This funding could then be matched through a variety of sources, including governments, collaborating partners, and through delivery of studies for industry.

## Recommendations

Based on the experiences of staff across test facilities, the following recommendations are made related to each factor (Table 4). These recommendations apply to both institutions targeting accreditation and prospective partners and funders of such projects.

**Table 4. Recommendations for institutions and collaborating partners seeking certification/accreditation for quality management systems.**

| Factor | Recommendations |
|---|---|
| Training programme | • Train the trainers–where regional expertise is lacking, assist key staff in accessing external training opportunities<br>• Emphasise practical training and on-the-job supervision for improved conformance with SOPs<br>• Regular all-staff generalised GLP/selected accreditation training |
| Project Planning | • Use all-facility meetings and working groups to plan, highlight success and identify risks/challenges<br>• Order activities in projects to allow issues to be addressed in parallel |
| Senior Leadership Support | • Engage with senior leadership on the benefits of GLP certification–including business opportunities and prestige—at the start of the project<br>• Include senior leadership support as a site-selection criterion |
| Infrastructure Development | • Manage construction projects internally–this can also boost the local economy<br>• Seek secondary investment as visible progress is made |
| Staff Structure | • Identify staffing risks, plan to employ more staff once more studies are attracted<br>• Provide extra support for SOP writing to technicians with high work loads |
| Procurement | • Prioritise local procurement, while balancing against risk of lower quality<br>• Identify sources of internal delay and simplify processes, consider use of SOPs<br>• Engage procurement and finance staff in GLP sensitisation training to promote understanding and smooth progress |
| Funding | • Funding flexibility to address unexpected costs which can emerge throughout<br>• Training in business management, marketing and networking skills to attract studies to sustain the facility and its GLP certification |
| Expertise and Support | • Identify and build on existing knowledge within non-GLP systems, including quality assurance and data management<br>• Consistent partner institution technical support for SOP development/implementation and IT data management systems<br>• Inter-facility learning, including site visits to view infrastructure, open source SOPs<br>• Identify opportunity to translate aspects of the quality management systems (e.g. SOPs) to other areas of the institution, to enhance research data quality more generally |

## Strengths and limitations

The strengths of this project are the diversity of participants involved in the study, capturing the views of staff in a diverse range of roles in five test facilities across three countries. This ensured that challenges and opportunities experienced by staff in all roles were reflected in the findings and ensured that the views of less often heard voices within research teams, such as those of technicians and administrators, were captured. This study is, however, limited by several factors. All facilities were engaged in insecticide testing, and findings may not be generally transferable beyond this specialty. However, while this may have influenced some factors such as external expertise and support, with the need for science-specific interpretation of the principles of GLP, for example, no identified factors are particularly science-specific and, therefore, we believe our findings are largely transferable to other contexts. The study also only includes the views and experiences of test facilities rather than including partners and stakeholders. This may mean that details related to some areas such as funding, expertise and support might not be included.

## Implications for collaborative research capacity strengthening

For partnership and consortia models of research capacity strengthening, these findings prompt reflection on roles and responsibilities within a research capacity strengthening project. In particular, that test facilities can use their own internal resources to address the identified high-control factors, that plans should allow additional time for interaction with external agencies to address medium-control factors, and that partners with access to expertise and funding should concentrate their efforts on supporting institutions to address low-control factors.

It also prompts reflection by funders, implementing agencies, and collaborating institutions on the intended end-goal of the research capacity strengthening initiative. If the objective is to include *independent sustainability*, the challenge becomes one of helping to bring these low control factors under greater control of the test facility. Projects must plan for financial sustainability robustly and comprehensively, invest in auxiliary laboratories and maintenance service providers, and consider broader regional capacity by investing in multiple test facilities to create a pool of individuals with relevant expertise who can drive future training and capacity building. This "capacity to build capacity" has been highlighted previously [31], and emphasises the benefit of approaching research capacity strengthening initiatives with an eye on national/international capacity, going beyond the institutional level, an approach which has been greatly underutilised to date. This may be particularly true of certifications such as GLP, because there are recurring costs associated with maintaining certification. Many of the challenges faced in in securing accreditation are likely to reoccur, particularly regarding staff turnover in key roles (quality assurance, data management) or maintaining infrastructure and equipment for continued GLP compliance.

For financial sustainability, projects must build in sustainability from the outset [31] and this research emphasises that if providing contract services forms all or part of this financial sustainability strategy, this must also include training in secondary skills such as business planning and marketing. However, the impact of an emphasis on financial stability on collaboration within networks must also be considered with the competition for studies by facilities within the network affecting the willingness of facilities to share best practice and collaborate. Within this project, training in business management skills has helped to improve the ability of test facilities to address issues related to funding. In addition, the development of regional expertise related to GLP through investment in individuals in multiple test facilities and facilitation of in-network capacity strengthening training, has helped form the beginnings of a

"pool" of individuals who could fulfil key roles or train others in the region. However, with test facilities citing a lack of confidence in implementing business plans, and with other regional capacity gaps related to service provision remaining, they cannot yet be considered to have fully addressed some of the low control factors that could affect their long term sustainability.

## Supporting information

**S1 File. Interview guide.**
(DOCX)

## Acknowledgments

We thank Jameel Bharmal, for his support in data collection and translation, and Russell Dacombe, for his technical input.

## Author Contributions

**Conceptualization:** Alex Wright, Imelda Bates.

**Data curation:** Sara Begg, Alex Wright.

**Formal analysis:** Sara Begg, Graham Small, Imelda Bates.

**Funding acquisition:** Graham Small, Sarah Moore, Ben Koudou, William Kisinza, Diabate Abdoulaye, Frank Mosha, Imelda Bates.

**Investigation:** Sara Begg, Graham Small, Imelda Bates.

**Methodology:** Sara Begg, Graham Small, Imelda Bates.

**Project administration:** Sara Begg, Alex Wright, Matt Kirby, Jason Moore, Robert Malima, Patrick Kija, Constant Edi.

**Resources:** Sarah Moore, Ben Koudou, William Kisinza, Diabate Abdoulaye, Frank Mosha.

**Supervision:** Graham Small, Imelda Bates.

**Validation:** Alex Wright, Matt Kirby, Sarah Moore, Ben Koudou, William Kisinza, Diabate Abdoulaye, Jason Moore, Robert Malima, Patrick Kija, Frank Mosha, Constant Edi.

**Visualization:** Sara Begg.

**Writing – original draft:** Sara Begg, Graham Small, Matt Kirby, Imelda Bates.

**Writing – review & editing:** Alex Wright, Sarah Moore, Ben Koudou, William Kisinza, Diabate Abdoulaye, Jason Moore, Robert Malima, Patrick Kija, Frank Mosha, Constant Edi.

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
