## [Decision Letter · Decision Letter 0]

6 Jul 2021

PONE-D-21-09716

Multi-site comparison of factors influencing progress of African insecticide testing facilities towards an international Quality Management System certification

PLOS ONE

Dear Dr. Begg,

Thank you for submitting your manuscript to PLOS ONE. After careful consideration, we feel that it has merit but does not fully meet PLOS ONE’s publication criteria as it currently stands. Therefore, we invite you to submit a revised version of the manuscript that addresses the points raised during the review process.

We look forward to receiving your revised manuscript.

Kind regards,

Ramzi Mansour

Academic Editor

PLOS ONE

Journal Requirements:

2. Please include your tables as part of your main manuscript and remove the individual files. Please note that supplementary tables should be uploaded as separate "supporting information" files.

3. Please include a copy of the interview guides used as a Supporting Information file.

4. This is a qualitative study , as such, we do not feel that any conclusions on the effectiveness of the test facilities using their  own internal resources to address high-control factors may be supported; thus, we ask that you revise the text (especially, but no limited to, the aims and Conclusions) to avoid unsupported statements.

5. During our internal checks, the in-house editorial staff noted that you conducted research or obtained samples in another country, specifically Burkina Faso. Please check the relevant national regulations and laws applying to foreign researchers and state whether you obtained the required permits and approvals. Please address this in your ethics statement in both the manuscript and submission information. In addition, please ensure that you have suitably acknowledged the contributions of any local collaborators involved in this work in your authorship list and/or Acknowledgements. Authorship criteria is based on the International Committee of Medical Journal Editors (ICMJE) Uniform Requirements for Manuscripts Submitted to Biomedical Journals - for further information please see here: https://journals.plos.org/plosone/s/authorship

6. We note that you have indicated that data from this study are available upon request. PLOS only allows data to be available upon request if there are legal or ethical restrictions on sharing data publicly. For information on unacceptable data access restrictions, please see http://journals.plos.org/plosone/s/data-availability#loc-unacceptable-data-access-restrictions.

Reviewers' comments:

Reviewer's Responses to Questions

**Comments to the Author**

1. Is the manuscript technically sound, and do the data support the conclusions?

Reviewer #1: Yes

2. Has the statistical analysis been performed appropriately and rigorously? 

Reviewer #1: N/A

3. Have the authors made all data underlying the findings in their manuscript fully available?

Reviewer #1: No

4. Is the manuscript presented in an intelligible fashion and written in standard English?

Reviewer #1: Yes

5. Review Comments to the Author

Reviewer #1: Excellent contribution, I am pleased to see this type of study in a serious journal. Withholding transcripts of interviews seems appropriate to me.

I would like to see more on the method for analysing the findings revealed through interviews, probably in section 2.3. Were the researchers looking for low, medium and high control from the start, or was that the criteria that emerged from the study? Any other references that you can add on that methodology of a largely qualitative approach? Also, please add a sentence about why you did not consider it worth running any statistical on the results, in 2.5. Please reword sentence line 546-547. Line 580 repeats so possibly say, Expanding on the observation noted above (or correct repetition). Typo line 574. Table 1, is IRSS private? explain if it is a private lab within a government entity.

ABSTRACT - talks about data for regulatory entities but this is never expanded on and who are 'these' is not clear. Is this a key point? If so, clarify better in the intro. At least describe the type of regulators (e.g. which Ministry). Or cite refs to say it is important. Otherwise, it seems a bit undeveloped.

6. PLOS authors have the option to publish the peer review history of their article (what does this mean?). If published, this will include your full peer review and any attached files.

Reviewer #1: **Yes: **M. Megan Quinlan

---

## [Author Response · Author response to Decision Letter 0]

18 Oct 2021

Editor comments

1. Please ensure that your manuscript meets PLOS ONE's style requirements 

This has been updated.

2. Please include your tables as part of your main manuscript and remove the individual files 

These have been added. 

3. Please include a copy of the interview guides used as a Supporting Information file. 

These have been added.

4. This is a qualitative study, as such, we do not feel that any conclusions on the effectiveness of the test facilities using their own internal resources to address high-control factors may be supported 

References to effectiveness have been removed, and simplified to clarify that test facilities can mobilise their own internal resources to address high control factors (without claiming that this work was effective):

“For partnership and consortia models of research capacity strengthening, test facilities can use their own internal resources to address identified high-control factors.”

5. During our internal checks, the in-house editorial staff noted that you conducted research or obtained samples in another country, specifically Burkina Faso. Please check the relevant national regulations and laws applying to foreign researchers and state whether you obtained the required permits and approvals. 

We have clarified that: “In line with LSTM’s guidance on Network and Capacity Strengthening Studies, senior management at the IHI (Tanzania) and IRSS (Burkina Faso) test facilities provided an institutional approval document to confirm willingness to participate and that the research did not require in-country approval.”

Please find text from this LSTM guidance here: 

In-Country REC Exemption Criteria

 1. Research on aspects of a Network of academics or professionals, where all participants are members of the Network 

 2. Capacity strengthening research, including facility assessments and interviews focussed on work-based matters (not personal)

Relevant collaborators from partner institutions are included as co-authors.

6. We note that you have indicated that data from this study are available upon request. PLOS only allows data to be available upon request if there are legal or ethical restrictions on sharing data publicly. 

While there are a relatively high number of interviewees, due to the variety in roles represented in these interviews we believe that transcripts of these interviews are still allow easy identification of individuals even when redacted, particularly in middle and senior management roles. 

We therefore feel it is not ethical to have these transcripts available as open access, for reasons including potential ramifications for employment or future funding when individuals speak critically about an individual, a process or a partnership.

In addition, interviews were conducted on the premise of anonymity, and this was the basis on which the study received approval from the LSTM REC.

Requests for access to this data may be made to ccr@lstmed.ac.uk or to rec@lstmed.ac.uk

Reviewer comments

1. I would like to see more on the method for analysing the findings revealed through interviews, probably in section 2.3. Were the researchers looking for low, medium and high control from the start, or was that the criteria that emerged from the study? Any other references that you can add on that methodology of a largely qualitative approach? 

We have clarified that the low, medium and high control criteria in the thematic framework were emergent findings, and that these were incorporated into the thematic framework to provide structure: 

“At this point, emergent themes related to the degree of control test facilities had over the factors affective progress were identified, and this was used to structure the thematic framework.”

Further detail across this section of the methods has been included, with particular reference to relevant references:

This study employed semi-structured interviews with staff at each of the five test facilities to construct a narrative of progress towards GLP compliance. A qualitative approach was used in order to capture the experiences and interpretation of those experiences by individuals in a wide range of roles (18). This ensured that there was scope to describe both expected and unexpected events or factors, and offer explanations for these events or factors in order to identify lessons for future capacity strengthening initiatives. In-person interviews were conducted at three test facilities, whilst remote interviews were conducted at two test facilities due to travel restrictions resulting from the COVID-19 pandemic.

And:

A framework analysis (20) was used to identify themes emerging from the interview transcripts following the five-step process of familiarization, identification of thematic framework, indexing, charting, and mapping/interpretation . The framework approach to analysis was employed to facilitate identification and understanding of the interactions and processes that influenced the rate of progress towards sustainable GLP certification and to facilitate comparison between test facilities, offering a systematic approach to manage qualitative data (20) that allows for both deductive and inductive approaches to analysis. An initial coding framework was based on findings from the case study exploring progress at the KCMUCo-PAMVERC Test Facility (17), with codes based on challenges and enablers to progress identified at this site. Transcripts of interviews conducted in this study were read in detail. Following this familiarization with the interview data, these codes were revised, and further codes were identified and incorporated into the framework. At this point, emergent themes related to the degree of control test facilities had over the factors affective progress were identified, and this was used to structure the thematic framework. 

This initially deductive approach, using findings from the KCMUCo-PAMVERC Test Facility to inform the initial framework, followed by an inductive approach to develop this framework, supported focus on the specific area of factors affecting GLP certification progress while allowing for unexpected aspects of the GLP certification process described by participants to be included in the analysis (18, 21), sharing some characteristics with Grounded Theory (22). All interview transcripts were indexed with these revised codes, using NVivo software (QSR International). To identify sections of the data that corresponded to the relevant theme, a narrative was constructed to define the key issues, find associations between those issues and, where possible, provide explanations. In sum, this facilitated the development of a model to describe this qualitative data, which can be tested in future research (23).

2. Also, please add a sentence about why you did not consider it worth running any statistical on the results, in 2.5.

 Added: “Statistical and quantitative methods were not used in this study as test facilities were at substantially different stages in the process of working towards GLP certification, limiting the usefulness of comparison of potentially relevant process-based quantitative data such as checklists and rendering comparison of outcome indicators such as efficiency or data quality not possible.”

3. Line 580 repeats so possibly say, Expanding on the observation noted above (or correct repetition). 

This is an illustrative quote for the text above (in italics), and this explains why it appears to be repetitive.

4. Please reword sentence line 546-547. 

This has been revised for clarity to: 

“As well as calibration and maintenance of equipment, test facilities required support from government bodies that oversaw waste management and animal welfare. As with calibration and maintenance of equipment, these were services and support systems that test facilities needed but had very little control over”

5. Typo line 574. 

Amended:

“Despite a good understanding of the business case for GLP and the need to attract studies from industry, test facilities generally felt they did not have the marketing skills necessary to sell their services on the open market, and identified this as a major risk to sustaining GLP certification:”

6. Table 1, is IRSS private? explain if it is a private lab within a government entity. 

Amended, this was an admin error:

“Government”

7. ABSTRACT - talks about data for regulatory entities but this is never expanded on and who are 'these' is not clear. Is this a key point? If so, clarify better in the intro. At least describe the type of regulators (e.g. which Ministry). Or cite refs to say it is important. Otherwise, it seems a bit undeveloped. 

Clarified that this is in reference to bodies that determine which products are appropriate for procurement for vector control globally 

“Data from these test facilities must be of a high quality to be accepted by regulatory authorities, including the WHO Prequalification Team for vector control products.”

---

## [Editor Report · Decision Letter 1]

28 Oct 2021

Multi-site comparison of factors influencing progress of African insecticide testing facilities towards an international Quality Management System certification

PONE-D-21-09716R1

Dear Dr. Begg,

We’re pleased to inform you that your manuscript has been judged scientifically suitable for publication and will be formally accepted for publication once it meets all outstanding technical requirements.

Kind regards,

Ramzi Mansour

Academic Editor

PLOS ONE

---

## [Editor Report · Acceptance letter]

4 Nov 2021

PONE-D-21-09716R1 

Multi-site comparison of factors influencing progress of African insecticide testing facilities towards an international Quality Management System certification 

Dear Dr. Begg:

I'm pleased to inform you that your manuscript has been deemed suitable for publication in PLOS ONE. Congratulations! Your manuscript is now with our production department. 

Kind regards, 

on behalf of

Dr. Ramzi Mansour 

Academic Editor

PLOS ONE